# Natural Antioxidant, Antibacterial, and Antiproliferative Activities of Ethanolic Extracts from *Punica granatum* L. Tree Barks Mediated by Extracellular Signal-Regulated Kinase

**DOI:** 10.3390/plants11172258

**Published:** 2022-08-30

**Authors:** Arpron Leesombun, Ladawan Sariya, Jarupha Taowan, Chowalit Nakthong, Orathai Thongjuy, Sookruetai Boonmasawai

**Affiliations:** 1Department of Pre-Clinic and Applied Animal Science, Faculty of Veterinary Science, Mahidol University, Nakhon Pathom 73170, Thailand; 2The Monitoring and Surveillance Center for Zoonotic Diseases in Wildlife and Exotic Animals (MoZWE), Faculty of Veterinary Science, Mahidol University, Nakhon Pathom 73170, Thailand; 3Department of Clinical Sciences and Public Health, Faculty of Veterinary Science, Mahidol University, Nakhon Pathom 73170, Thailand; 4The Center of Veterinary Diagnosis, Faculty of Veterinary Science, Mahidol University, Nakhon Pathom 73170, Thailand

**Keywords:** pomegranate tree barks, pomegranate fruit peels, anticancer, HeLa cells, HepG2 cells, ERK1/2 expression

## Abstract

The nonedible parts of the pomegranate plant, such as tree barks and fruit peels, have pharmacological properties that are useful in traditional medicine. To increase their value, this study aimed to compare the antioxidative and antibacterial effects of ethanolic extracts from pomegranate barks (PBE) and peels (PPE). The antiproliferative effects on HeLa and HepG2 cells through the extracellular signal-regulated kinase pathway were also evaluated. The results indicated that the total amounts of phenolics and flavonoids of PBE and PPE were 574.64 and 242.60 mg equivalent gallic acid/g sample and 52.98 and 23.08 mg equivalent quercetin/g sample, respectively. Gas chromatography–mass spectrometry revealed that 5-hdroxymethylfurfural was the major component of both PBE (23.76%) and PPE (33.19%). The 2,2-diphenyl-1-picryl-hydrazyl-hydrate free radical scavenging capacities of PBE and PPE, in terms of the IC_50_ value, were 4.1 and 9.6 µg/mL, respectively. PBE had a greater potent antibacterial effect against *Escherichia coli*, *Staphylococcus aureus*, *Salmonella* Enteritidis, and *S.* Typhimurium. PBE and PPE (1000 µg/mL) had exhibited no cytotoxic effects on LLC-MK2. PBE and PPE (250 and 1000 µg/mL, respectively) treatments were safe for BHK-21. Both extracts significantly inhibited HepG2 and HeLa cell proliferations at 10 and 50 µg/mL, respectively (*p* < 0.001). The results indicated that PBE and PPE have remarkable efficiencies as free radical scavengers and antibacterial agents, with PBE exhibiting greater efficiency. The inhibitory effects on HepG2 might be through the modulation of the ERK1/2 expression. PBE and PPE have the potential for use as optional supplementary antioxidative, antibacterial, and anticancer agents.

## 1. Introduction

Cancer is a significant public health concern and is among the leading causes of death worldwide. By the end of 2022, it is estimated that 1.9 million new cancer cases will be diagnosed in the United States, which will lead to 609,360 deaths [1]. In particular, cervical and liver cancers have high global prevalence and mortality rates. Cervical cancer poses a threat to women’s health; moreover, it is prevalent in developed countries. Recently, there have been reports on human papillomavirus infections related to cancer [2,3,4]. On the other hand, liver cancer is the sixth most common cancer worldwide. The etiology of chronic hepatitis B and C is mainly infection. Obesity, fatty liver disease, cirrhosis, and diabetes are also risk factors [5].

At present, treatments for cancer include surgery, chemotherapy, and radiotherapy. However, these conventional chemotherapies have serious side effects and are sometimes inefficient [6]. Thus, new strategies to control the impacts on human health need to be developed. In traditional medicine, the use of some plants for the treatment of cancer has been proposed, such as *Catharanthus roseus*, *Viscum album*, *Taxus baccata*, *Camptotheca acuminata*, and *Phaleria macrocarpa* [6,7,8,9]. Plant extracts have beneficial characteristic, e.g., they have a variety of pharmacological effects, are easily available, and are less toxic. Recently, many studies have evaluated the effects of highly potent herbal-based compounds on cancer treatments [6].

Pomegranate (*Punica granatum* L.), an ancient fruit belonging to the Lythraceae family, is one of the important commercial horticultural crops in Asian countries and the whole Mediterranean region. Its fruits, leaves, roots, or flowers, which are commonly used in traditional medicine, contain several phytochemical constituents, especially phenolic compounds with various biological properties, including antiproliferative, antiinvasive, antimetastatic, and apoptotic properties, against cancers [10,11,12]. Most studies relate to the pharmacological effects of pomegranate concerning the juice or the whole fruit [11]. Nonedible parts, such as fruit peels, by-products after fruit juice processing, and tree barks, may have similar pharmacological properties to those found in the fruit, which can increase their value.

In the last few years, scientific investigations into pomegranate peel extracts have gained prominence [13]. Pomegranate peels contain phenolic compounds, such as flavonoids (anthocyanins and catechins), tannins (ellagitannins and ellagic acid derivatives: punicalagin, punicalin, and pedunculagin), and phenolic acids (hydroxycinnamic and hydroxybenzoic acids), which provide many health benefits [12,14,15]. Many studies have demonstrated the antioxidant and antibacterial capacities of PPE for nutraceutical application [16,17,18,19,20]. Although a pomegranate bark is used in traditional medicine to alleviate diarrhea, studies on pomegranate bark extracts are scarce. The major constituents of pomegranate bark are ellagitannins, piperidine alkaloids, pyrrolidine alkaloids, and pelletierine alkaloids [21], which have molluscicidal activity against the snail *Lymnaea acuminate* [22] and anti-candida activities [23].

This study aimed to explore the free radical scavenging activity, antibacterial, and antiproliferative effects of the ethanolic extracts of pomegranate barks (PBE) compared with those of pomegranate peels (PPE). The mechanisms of antiproliferative effects through the extracellular signal-regulated kinase (ERK) signaling pathway on human cervical and hepatic cancers of both plant extracts were also investigated. Knowledge on the comparative pharmacological activities of PBE and PPE may expand the substitute treatment options from these two natural resources.

## 2. Results and Discussion

### 2.1. Total Phenolic Content, Flavonoid Content, and free radical scavenging activity

Phenolic and flavonoid contents were found to be two times higher in PBE than in PPE (Table 1). The 2,2-diphenyl-1-picryl-hydrazyl-hydrate (DPPH^•^) free radical scavenging capacity, in terms of the half maximal inhibitory concentration (IC_50_) value, of PBE was lower than those of vitamin C, vitamin E, and butylated hydroxytoluene (BHT) (4.90, 11.50, and 109.30 µg/mL, respectively), whereas that of PPE was lower than those of vitamin E and BHT. Therefore, PBE was found to exhibit higher DPPH^•^ free radical scavenging activity than PPE, vitamin C, vitamin E, and BHT.

Polyphenols are the major secondary metabolites of most plants with high antioxidant capacities [24]. Polyphenols are composed of various compounds that were classified according to their chemical structure as flavonoids, phenolic acids, stilbenes, and lignans [25,26]. Their scavenging activities on reactive radicals depend on the number and position of the hydroxyl groups present in their molecules [27]. Pomegranate tree barks and fruit peels contain high amounts of polyphenols, which provide many health benefits, including antioxidant activity [28]. The pomegranate peel extract using 80% ethanol at 800 ppm exhibited higher antioxidant activity than BHT [29].

In this study, PBE was found to have two times higher phenolic content than PPE. Fazio et al. reported the lower level of phenolic level of pomegranate peels extracted by acetone and methanol (186 ± 8.7 and 178.7 ± 2.5 mg GAE/g of dry weight, respectively) [30]. The total phenolic content of PPE in this study was similar to those reported by Hasnaoui et al. (205–276 mg GAE/g) [31], Benchagra et al. (283.86 mg GAE/g) [32] and Mastrogiovanni et al. (202.22 mg GAE/g) [33] but lower than that reported by Keta et al. (367.33 mg GAE/g) [34]. On the other hand, its flavonoid content was higher than that reported by Keta et al. (6.06 mg catechin/g) [34] but lower than that reported by Benchagra et al. (185.37 mg QE/g) [32]. Polyphenols were detected from PBE and PPE with varying levels of 0.002 mg to more than 1 g [28,35,36]. Phenols, terpenes, alkanes, alkaloids, esters, steroids, and acids were detected in the extracts of each part of the pomegranate [37]. The phenolic compounds in previous studies have varying levels depending on the cultivar, plant processing technique, drying strategies, and different extraction protocols [35,36,38]. The PBE extracted using ethanol had higher contents of total phenolic and flavonoid compounds than those extracted using water, chloroform, acetone, and methanol [39]. Furthermore, the ethanolic and methanolic extracts of pomegranate peels were found to contain higher total phenolic compounds. The concentrations of the total phenolic and total flavonoid compounds were proportional to the antioxidant values [40].

### 2.2. Chemical Composition

The yields of PBE and PPE based on their dry weights were 0.96% and 1.43%, respectively. Figure 1 and Figure 2 present the chromatogram of the main components of PBE and PPE, respectively. A total of 10 compounds of PBE and PPE were determined via gas chromatography–mass spectrometry analysis (GC–MS) (Table 2 and Table 3). 5-Hydroxymethylfurfural (5HMF) is a major component of both PBE (23.76%) and PPE (33.19%). 5HMF, a high potential chemical derived from real biomass, could be directly produced by an acid catalyzed reaction of sugars, mainly hexoses, which are naturally abundant in many edible plants [41]. 5HMF which is widely contained in various plants, possess several pharmacological properties, such as antioxidant and antiproliferative effects on cancer cells [42], antibiofilm activity on Gram-positive [43] and Gram-negative bacteria [44,45], neuroprotective effects [46], and anti-inflammatory activity [47]. Natural 5HMF could be found in pomegranate juices (27.32 mg/kg). 5HMF formation in pomegranate juices has been shown to increase at a temperature above 200 °C and pH value above 7 [48]. The study by Yassin et al. demonstrated that the main antibacterial ingredients found in the methanolic extract of fruit peels were 5HMF (37.55%), octadecanoic acid (16.89%), and furfural (14.62%). On the other hand, the acetonic extracts contained 5HMF (28.41%), furfural (11.29%), and 2-furancarboxaldehyde, 5-methyl (9.58%). Hexadecanoic acid, methyl ester (1.86%) was also found in the acetonic extracts [49]. Furfural found in pomegranate peels had been reported on antibacterial effects [50]. Moreover, the hydromethanolic extract of *P. granatum* var. nana fruit containing 5HMF (37%) exhibited antibacterial effects against plant pathogenic microorganisms *Erwinia amylovora* and *E. vitivora* [51]. In the study by Hanafy et al., the 5HMF concentrations of the methanolic and ethanolic extracts of fruit peels reached 65.78% and 48.43%, respectively, and exhibited high activity as antioxidant [40]. This indicated that 5HMF is the active compound that processed the antioxidant and antibacterial activities. However, 5HMF and other polyphenols contained in PBE and PPE might theoretically work together because the combinations of polyphenols that have similar mechanisms of action could exert synergistic antioxidative effects. One example was the combination of curcumin and resveratrol which exhibit a higher DPPH^•^ free radical scavenging activity than the individual constituent [52]. Thus, the synergistic effects of such a combination in the PBE and PPE on the bioactivities should be investigated in the future.

### 2.3. Antibacterial Effect

The minimum inhibitory concentration (MIC) values of PBE on *E. coli*, *S. aureus*, *S.* Enteritidis, and *S*. Typhimurium were 6.25, 1.56, 1.56, and 3.125 mg/mL, whereas for PPE, the MIC values were 25, 6.25, 3.125, and 3.125 mg/mL, respectively (Table 4). The results indicated that the MIC values of PBE were lower than those of PPE on *E. coli*, *S. aureus*, and *S.* Enteritidis. Therefore, PBE was found to exhibit strong antibacterial activity against the bacteria reference strains as well as Gram-positive and Gram-negative pathogens.

Several studies on the antibacterial activities of PPE against Gram-positive and Gram-negative pathogens have been conducted but none have been carried out on the antibacterial activities of PBE. Our study first demonstrated that both PBE and PPE can inhibit the growth of the major foodborne bacteria, *E. coli*, *S. aureus*, and *S. enterica* (serovar Enteritidis and Typhimurium) [53]. The MIC values of PBE on *E. coli*, *S. aureus*, *S.* Enteritidis, and *S.* Typhimurium were 6.25, 1.56, 1.56, and 3.125 mg/mL, whereas those of PPE were 25, 6.25, 3.125, and 3.125 mg/mL, respectively. The results indicated that the MIC values of PBE were lower than those of PPE on *E. coli*, *S. aureus*, and *S.* Enteritidis. Moreover, PPE was found to exhibit strong antibacterial activity against the bacteria reference strains as well as Gram-positive and Gram-negative pathogens. Our results agreed with those of previous studies that pomegranate peel was more effective against species, *S. aureus* (MIC = 0.2–15.63 mg/mL), than against *E. coli* (MIC = 6.4–125 mg/mL) [14,49,54]. The MIC values of pomegranate peel extracts exhibited an antimicrobial activity against *Salmonella* spp. at various concentrations, from 0.06 to 150 mg/mL. Pomegranate peel extracts at 10.75–12.75 mg/mL exerted bactericidal effects against *S.* Enteritidis isolated from chicken meat [53,55,56].

Polyphenol constituents, including tannins, catechins, and gallic and ellagic acid, mainly exert antibacterial effects against *S. aureus* and *E. coli* [57]. Polyphenols exerted antibacterialeffect through various mechanisms. The binding of polyphenols to important bacterial protein, such as topoisomerases of *Mycobacterium* spp. [58], sortase A of *Streptococcus mutans* membrane [59], and porins at the outer membrane of Gram-negative bacteria [60,61], can change the bacterial structure and crucial activity of bacteria. Some active flavonoids could destroy the synthesis of bacterial RNA [62] and DNA [63]. The bacterial cell wall, outer membrane, and their components, such as peptidoglycan and lipopolysaccharide, can be directly destabilizing by polyphenols [64,65]. Some polyphenols, such as resveratrol, piceatannol, quercetin, quercetrin, and quercetin-3-β-D glucoside, act as ATPase inhibitors on Gram-negative bacteria [66]. The anti-quorum sensing activity of polyphenols, such as curcumin [67] and quercetin [68], demonstrated the biofilm inhibition of pathogens, such as *E. coli*, *P. aeruginosa*, and *Proteus mirabilis*. The antimicrobial activity of 5HMF has been reported on Gram-positive and Gram-negative bacteria [43,45,69]. The antibacterial mechanism of 5HMF was not clearly known, although there are reports to prove that 5HMF inhibits biofilm formation by targeting the quorum-sensing system. It was found to be effective in the initial stage of biofilm formation. Moreover, 5HMF treatment inhibited the production of extracellular polymeric substances of bacteria, including polysaccharide and protein production [45]. These confirm that the antibacterial effects of PPE are related to the action of polyphenolic compounds [70]. Owing to their antioxidative and antimicrobial properties, PBE and PPE have a high potential to improve product characteristics and inhibit bacterial colonization during food industrial processes. Thus, PBE and PPE might be used as natural sources of preservative agents instead of BHT to extend the shelf-life of food products [71].

### 2.4. The Cytotoxic Effects of PBE and PPE on LLC-MK2 and BHK-21 Cells

PBE at 1000 µg/mL did not inhibit rhesus monkey kidney epithelial cell (LLC-MK2) proliferation, whereas PBE at ≥500 µg/mL significantly inhibited baby hamster kidney cell (BHK-21) proliferation compared with Dulbecco’s Modified Eagle’s Medium (DMEM) treatment (Figure 3a,b). The PPE treatments on LLC-MK2 and BHK-21 cells at 1000 µg/mL exerted nonsignificant cytotoxic effects when compared with the DMEM treatment group via 3-(4,5-dimethyl-2-thiazolyl)-2,5-diphenyl-2H-tetrazolium bromide (MTT) and sulforhodamine B (SRB) assay (Figure 4a,b). Consistent with the study by Sineh et al., PPE from *P. granatum* L. var. spinosa at 600 µg/mL did not significantly inhibit L929 mouse fibroblasts [72]. This finding indicated the high level of safety of PBE and PPE for cancer treatment.

### 2.5. Antiproliferative Effects of PBE and PPE on HeLa and HepG2 Cells

A similar result was obtained for the PBE and PPE treatments. The inhibitory effects of PBE started at 50 µg/mL on cervical cancer cells (Figure 3c), and a significant antiproliferative effect was observed on human liver hepatocellular carcinoma (HepG2) cells at 10 µg/mL (Figure 3d). PPE at 50 µg/mL significantly inhibited human cervical adenocarcinoma (HeLa) cell proliferation compared with the control (Figure 4c). The PPE treatments initially exerted inhibitory effects on HepG2 growth at 10 µg/mL (Figure 4d). Furthermore, the antiproliferative effects of both extracts on HepG2 cells occurred in a dose-dependent manner (Figure 3d and Figure 4d). Based on the IC_50_ values of the extracts on HeLa and HepG2 cells, similar results were obtained between the MTT and SRB assays, and HepG2 cells were more sensitive to the extracts compared with HeLa cells (Table 5).

### 2.6. Immunoblotting

To evaluate the signaling mechanism of PBE and PPE on cancer cell proliferations, immunoblotting was performed to observe the total ERK1/2 protein expression in LLC-MK2, BHK-21, HeLa, and HepG2 cells after 24 h incubation with PBE and PPE (Figure 5). The PBE and PPE treatments on HepG2 at 10 µg/mL significantly decreased the ERK1/2 protein expression (0.49 and 0.47 times, respectively) when compared with the DMEM treatment. In HeLa cells, the PBE and PPE treatments at 50 µg/mL could significantly inhibit cancer cell proliferation (Figure 3c and Figure 4c); however, reduction in ERK1/2 protein (0.91 and 0.8 times compared with DMEM, respectively) was not observed.

The study of antiproliferative effects on HepG2 demonstrated that the signaling mechanism of PBE and PPE might be exerted by a reduction in total ERK1/2 protein expression. The Ras/Raf/MEK/ERK signaling pathway regulates the sequential processes of hepatocellular carcinoma proliferation [73]. ERK1/2 is one of four members in the ERK family (JNK1/2/3, ERK1/2, ERK5, and p38 MAPK) and the downstream protein target of MEK1/2 phosphorylation [74]. An optimal ERK signaling is required for cancer cell growth, and a much higher ERK1/2 expression is related to the drug resistance of cancer [75]. At present, ERK inhibitors, such as ulixertinib, SCH772964, and LY3214996, are used to improve therapeutic outcomes [76]. Furthermore, quercetin, the flavonoid identified from pomegranate [77], exerts inhibitory effects on HepG2 proliferation through the reduction in ERK1/2 protein [78]. Thus, the PBE and PPE had the potential to inhibit HepG2 proliferation through the early steps of the Ras/Raf/MEK/ERK signaling pathway.

In HeLa cells, the active form of downstream Ras could induce ERK1/2 activation, leading to increased expression of Noxa, the Bcl-2 homology 3 (BH3) protein, and step into cancer progressive cascades [78]. Antiproliferation effects on HeLa cells could be induced by reducing ERK1/2 protein after treatments with kaempferol (5 μM) and quercetin (1 μM) [79]. A previous study demonstrated that the proliferation of HeLa cells is related to multiple signaling pathways, including PDGF/AKT [80], JAK/STAT [81], and ROS-MAPK-mediated mitochondrial signaling pathways [82].

The previous study showed that 5HMF exhibited the dual effects on carcinogenesis as inducer and suppressor. Conversely, some reports demonstrated induction effects of sulfooxymethyl and chloromethyl derivatives of 5HMF on skin papilloma, hepatic tumor [83], and colonic aberrant crypt foci in mice [84]. Contrarily, 5HMF could inhibit cancer cell proliferation by inducing apoptosis through G0/G1 arrest; it could also inhibit the reactive oxygen species-mediated signal transduction pathway in melanoma [42]. 5HMF exhibited an indirect activity related to the MAPK signaling pathway. This active constituent could exert the anti-inflammatory activity by inhibiting Fc epsilon receptor (FcεRI)-mediated phosphorylation of ERK1/2 [85]. Moreover, c-Jun N-terminal kinase (JNK), IκBα, NF-κB p65, the mammalian target of rapamycin (mTOR) and protein kinase B (Akt) were target of 5HMF [47]. The anticancer effects and mechanisms of 5HMF and other constituents of PBE and PPE on HepG2 and Hela are interesting and must be studied further.

## 3. Materials and Methods

### 3.1. Chemicals and Reagents

The Folin–Ciocalteu reagent was obtained from Merck (Darmstadt, Germany). DPPH^•^ reagent, sodium carbonate, gallic acid, sodium nitrite, aluminum chloride, sodium hydroxide, quercetin, vitamin C (ascorbic acid), vitamin E (dl-alpha tocopherol), BHT, dimethyl sulfoxide (DMSO), DMEM, fetal bovine serum (FBS), PD-98059, 5-fluorouracil, MTT, SRB, trichloroacetic acid, acetic acid, 3,3′-Diaminobenzidine (DAB) substrate kit, resazurin, amikacin, ceftriaxone, and ciprofloxacin were all obtained from Sigma Aldrich (St. Louis, MO, USA). Sheep’s blood and Muller–Hinton broth were from Oxoid (Hampshire, United Kingdom). Bio-Rad protein assay kit was from Life Science (Bangkok, Thailand), and anti-p44/42 MAPK (ERK1/2) primary monoclonal antibody, anti-beta-actin and anti-rabbit IgG-conjugated horseradish peroxidase were from Cell Signaling (Danvers, MA, USA).

### 3.2. Plant Extracts Preparations

Pomegranate (*Punica granatum* L.) tree barks and fruit peels used in this study were obtained from the central region of Thailand. The plant was identified and deposited in the herbarium of the Faculty of Pharmacy, Mahidol University, Thailand. The voucher specimen was PBM-005493-4. The tree barks and fruit peels were oven-dried at 60 °C for 48 h before being ground into coarse powder using High-Speed Grinder PG2500 (Spring Green Evolution, Samut Prakan, Thailand) at 3200 rpm for 5 s. The ground plant materials were soaked into 95% ethanol (1000 g: 1.5 L) and incubated in a dark cabinet at room temperature for 3-day period. The solvent was filtered through sterile gauze and evaporated using the Büchi R-205 Rotary Evaporator with V-800 Vacuum Pump at 40 °C and 175-mbar pressure (BÜCHI, Flawil, Switzerland). Then, 100 g of extract was lyophilized in a Labconco FreeZone 4.5 L Freeze Dryer with Lyo-Works™ Operating System (Labconco, Kansas City, MO, USA) and stored at −20 °C until use. The yields of the extracts were calculated in percent yield based on the dry weights of tree barks and fruit peels.

### 3.3. Determination of Total Phenolic Content

Total phenolic content was determined using the Folin–Ciocalteu method, with some modifications [28]. The plant extract solution was prepared by mixing 500 µL of deionized water and 125 µL of the stock extract solution (1000 μg/mL). Subsequently, 125 µL of Folin–Ciocalteu reagent was added and allowed to react for 5 min. Then, 400 µL of 7.5% sodium carbonate solution was added, and the mixture was incubated for 30 min at room temperature. The absorbance was read at a wavelength of 760 nm using Synergy H1 Hybrid Multi-Mode Microplate Reader (BIOTEK, Winooski, VT, USA). Gallic acid was used to prepare the standard curve. The range for the calibration was 40–240 μg/mL. The gallic acid solutions and the results were expressed in GAE/g crude extract.

### 3.4. Determination of Flavonoid Content

The flavonoid content was determined using a modified aluminum chloride colorimetric method, with some modifications [86]. In brief, 250 µL of the extract solution (1000 μg/mL) was mixed with 1.25 mL deionized water. The 5% sodium nitrite solution (75 µL) was added, and the mixture was allowed to stand for 5 min. Then, 150 µL of 10% aluminum chloride was added, followed by 500 µL of 1 M sodium hydroxide; the mixture was diluted with 275 µL of deionized water and allowed to stand for 6 min. The absorbance of the mixture was immediately measured at a wavelength of 510 nm using Synergy H1 Hybrid Multi-Mode Microplate Reader (BIOTEK, Winooski, VT, USA). The calibration curves were made by preparing various concentrations of 30–300 μg/mL quercetin solution.

### 3.5. Determination of DPPH^•^ Free Radical Scavenging Activity

The antioxidant capacity of the extracts was determined using the DPPH^•^ method, with some modifications [48]. Vitamin C, vitamin E, and BHT were used as positive controls. Various concentrations of PBE and PPE (1–1000 µg/mL in ethanol) were mixed with DPPH^•^ solution (ratio, 1:1) and incubated in the dark at room temperature for 30 min. The absorbance of the mixture was then measured at a wavelength of 517 nm using Synergy HT Multi-Detection Microplate Reader (BIOTEK, Winooski, VT, USA). The free radical scavenging activity was calculated using the following equation: DPPH^•^ antiradical scavenging capacity (%) = [(A blank − A sample)/A blank] × 100. Blank is the absorbance value of the control reaction, and A sample is the absorbance value of each extract. The IC_50_ values represent the concentration of the sample required to scavenge 50% of DPPH^•^ radicals.

### 3.6. Gas Chromatography-Mass Spectrometry

The chemical composition of the PBE and PPE was analyzed by GC–MS. The stock samples (1 mg/mL) were diluted with methanol (1–10 µg/mL) and then injected in the split mode (1:10 split ratio) into the GC–MS model Agilent 7890A/5977B GC/MSD system with 19091S-433 capillary column (0.25 µm film thickness × 0.25 mm in diameter × 30 m in length) (Agilent Tech., Santa Clara, CA, USA) at a flow rate of 1 mL·min^−1^ in carrier gas: helium and injector temperature of 250 °C. The oven was set to 70 °C for 5 min and ramped to 250 °C at 10 °C/min, with a final hold of 5 min. The ion source temperature of MS conditions was 230 °C, including an ionization energy of 70 eV, and a mass scan range of 35–550 *m*/*z*. Compounds were identified by matching GC–MS results with the retention time and spectral database of the NIST library.

### 3.7. Determination of Minimum Inhibitory Concentration of PPE and PBE

The broth microdilution method according to the CLSI guidelines [87] was employed to determine the antimicrobial effects of PBE and PPE on *E. coli* ATCC^®^25922, *S. aureus* ATCC^®^29213, *S.* Enteritidis ATCC^®^13076, and *S.* Typhimurium ATCC^®^14028. The bacteria colonies from a fresh overnight culture on tryptic soy agar plates (Merck, Darmstadt, Germany) containing 5% sheep’s blood were suspended in 0.85% sterile saline to obtain a 0.5 McFarland standard and then diluted with Muller–Hinton broth. The PBE and PPE subjected to two-fold serial dilutions starting from 100 mg/mL were added into the triplicate wells of a sterile 96-well plate and incubated with inoculum bacteria (final concentration = 1–5 × 10^5^ colony forming units/mL) at 37 °C for 16–20 h. Then, 0.015% resazurin solution was added (30 µL/well), and further incubated for 2–4 h. The lowest concentration of the plant extracts without any color change of blue resazurin was recorded as the MIC [88]. To verify the sterility of the procedure, amikacin, ceftriaxone, and ciprofloxacin were used as positive controls.

### 3.8. Cell Cultures and Treatments

LLC-MK2, BHK-21, HeLa, and HepG2 cells were obtained from the Monitoring and Surveillance Center for Zoonotic Diseases in Wildlife and Exotic Animals (MoZWE), Faculty of Veterinary Science, Mahidol University, Thailand. The cells were cultured in DMEM supplemented with 10% FBS, 100 unit/mL penicillin G sodium, and 100 µg/mL streptomycin at 37 °C in a 5% CO_2_ atmosphere. After the cells reached confluence, they were trypsinized, and cell suspensions were prepared with culture media (5 × 10^4^ cells/mL). The cell suspensions (100 µL/well) were cultured in 96-well plates and incubated at 37 °C in a 5% CO_2_ atmosphere for 24 h before treatments in triplicate wells with various concentrations of PBE and PPE (1–1000 µg/mL) dissolved in 0.001% DMSO. The mitogen activated protein kinase inhibitor PD-98059 (20 µM), 5-fluorouracil (5-FU) (20 µg/mL), and DMEM were used as the controls. After 24 h incubation, the cytotoxicity and antiproliferative activity were evaluated by using MTT and SRB assays.

### 3.9. Evaluation of Cytotoxicity and Antiproliferative Activity of PPE and PBE

MTT assay was modified from Mosmann’s method [89]. After cell treatments, 10 µL of MTT (2 mg/mL) was added into 96-well plates and incubated at 37 °C for 3 h. The intracellular formazan occurring from a reduction in dye was solubilized by 200 µL DMSO. Then, a quantity of formazan that was directly proportional to the viable cell number was determined using Synergy HT Multi-Detection Microplate Reader (BIOTEK, Winooski, VT, USA) at a wavelength of 570 nm.

Cell proliferation was also evaluated via SRB calorimetric assay [90]. After the incubation period, 50 µL ice-cold 50% trichloroacetic acid in sterile distilled water was incubated with the cells at 4 °C for 1 h. The fixed cells in plates were gently washed 5 times with distilled water and dried at room temperature. The epithelial cells and cancer cells were stained with 50 µL SRB solution (0.4% *w*/*v* SRB dissolved in 1% acetic acid) for 30 min. Then, 1% acetic acid was used for 3–5 washes, and the plates were left to air dry. The culture plates were shaken on a plate shaker for 5 min at room temperature. Furthermore, the SRB dye in the cells was solubilized with 10 mM un-buffered Tris-based solution (pH 10.5) 200 µL/well, and its absorbance measured at a wavelength of 540 nm. The antiproliferative activity was calculated as the percent cell proliferation, expressed as [(Absorbance of treatment/Absorbance of control)] × 100. All expressed data were calculated from triplicate wells in each treatment.

### 3.10. Immunoblotting

After a 24-h incubation, the treated cells were washed with cold phosphate buffer saline and then lysed in cold extract buffer (50 mM Tris base; 10 mM EDTA; 1% (*v*/*v*) Triton X-100; 0.57 mM PMSF; 1.5 µM pepstatin A; 2 µM leupeptin). The Bio-Rad protein assay kit was used to determine protein concentrations. Equal amounts of protein samples were loaded into each lane of a 12% separating SDS/PAGE gel. After the gels were electrophoresed for 2 h at a constant voltage of 120 V, the separated protein bands were transferred onto a nitrocellulose membrane. The membranes were blocked in 5% low-fat milk in Tris-buffered saline and incubated with primary monoclonal antibody for 90 min (anti-p44/42 MAPK (ERK1/2) [dilution 1:500], anti-beta-actin [dilution 1:5000]) followed by incubation with anti-rabbit IgG-conjugated horseradish peroxidase (dilution 1:2000) (Cell Signaling, USA) for 60 min. The p44/42 MAPK (ERK1/2) antibody was used to detect endogenous levels of ERK1/2 protein. Immunoblots were developed with a DAB substrate kit.

### 3.11. Statistical Analysis

All statistical analyses were conducted using SPSS version 21.0. The IC_50_ value was calculated in GraphPad Prism version 6.0. The ImageJ program was used to determine the density of the protein band on blotting paper. The mean values of the data among treatment groups were compared using one-way analysis of variance and the *t*-test. Differences with *p*-values < 0.05 were regarded as statistically significant.

## 4. Conclusions

The results suggested that the ethanolic extracts from pomegranate tree barks had higher phenolic contents, stronger free radical scavenging capacity, and higher potent antibacterial activity than the fruit peel extracts. Furthermore, PBE and PPE were found to be safe and have comparable bioactivities on human hepatocellular carcinoma and cervical cancer treatments. The signaling mechanism of the antiproliferative effects of PBE and PPE on human hepatocellular carcinoma might be exerted by reducing the ERK1/2 expression. This study demonstrates the pharmacological potential of pomegranate tree barks for use as supplemental agents to improve human health.

## Figures and Tables

**Figure 1 plants-11-02258-f001:**
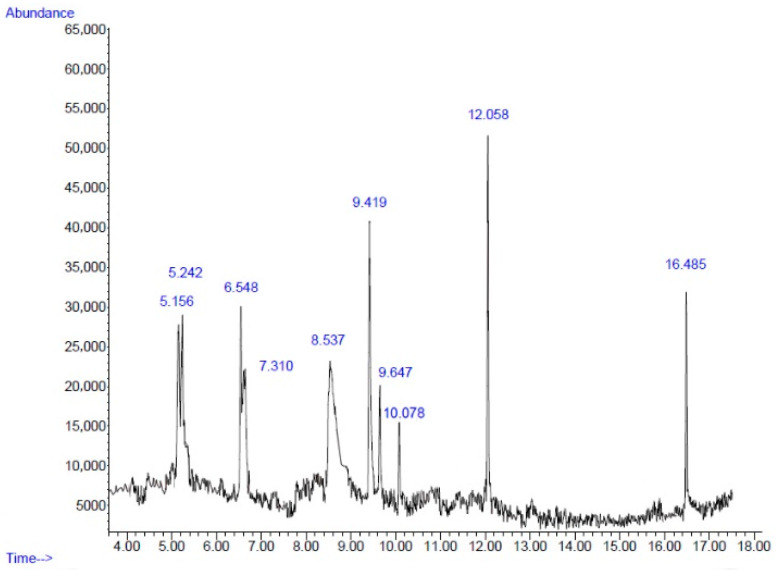
Chromatogram of main components of the ethanolic extracts from pomegranate barks obtained via GC-MS.

**Figure 2 plants-11-02258-f002:**
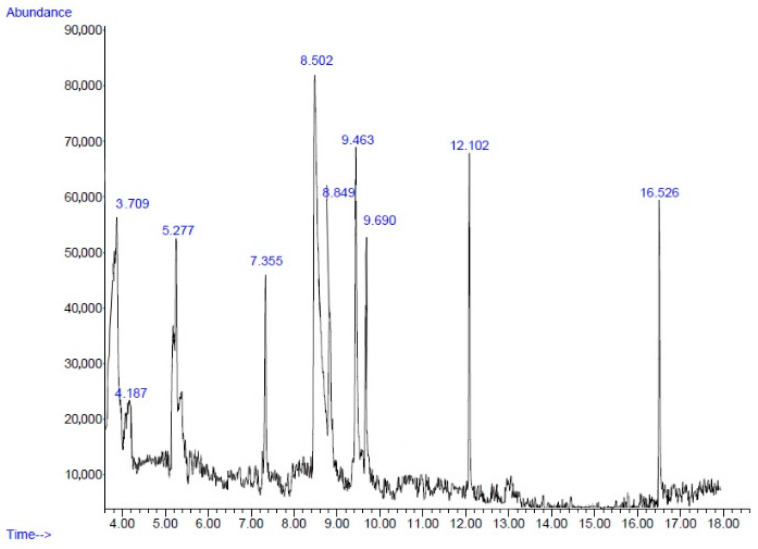
Chromatogram of main components of the ethanolic extracts from pomegranate peels obtained via GC-MS.

**Figure 3 plants-11-02258-f003:**
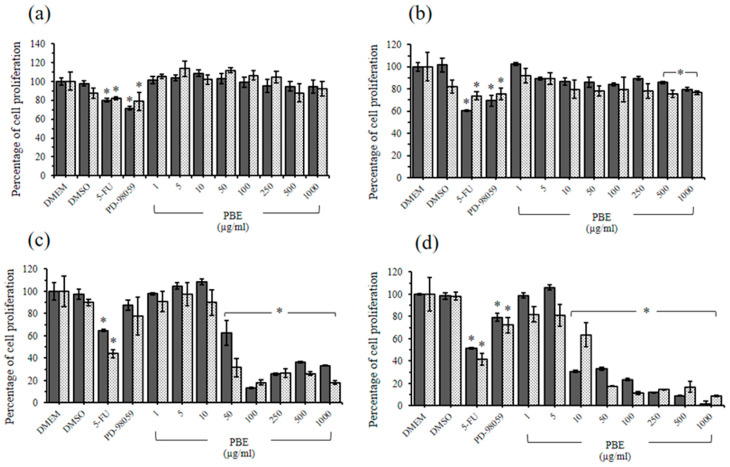
Cell proliferation following treatments with ethanolic extracts from pomegranate barks (PBE). The LLC-MK2 (**a**), BHK-21 (**b**), HeLa (**c**), and HepG2 (**d**) cell proliferations following treatments with PBE for 24 h via MTT (dark shade bar) and SRB (light shade bar) assays. Each bar represents the mean ± SD of three experiments per group. * The differences between DMEM and treatments were significant (*p* < 0.001).

**Figure 4 plants-11-02258-f004:**
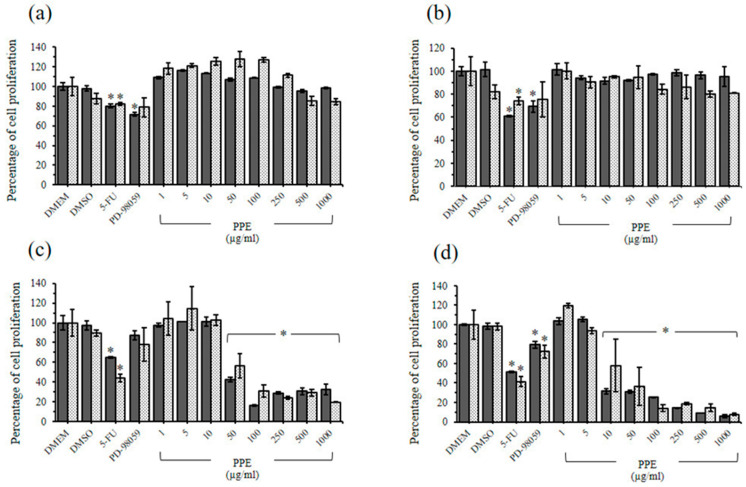
Cell proliferation following treatments with ethanolic extracts from pomegranate peels (PPE). The LLC-MK2 (**a**), BHK-21 (**b**), HeLa (**c**), and HepG2 (**d**) cell proliferations following treatments with PPE for 24 h via MTT (dark shade bar) and SRB (light shade bar) assays. Each bar represents the mean ± SD of three experiments per group. * The differences between DMEM and treatments were significant (*p* < 0.001).

**Figure 5 plants-11-02258-f005:**
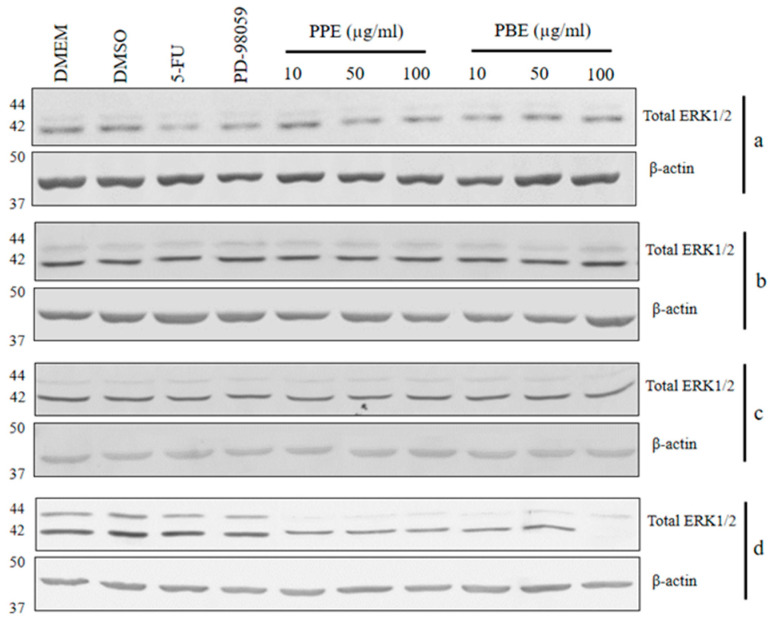
The ERK1/2 protein expression in LLC-MK2 (**a**), BHK-21 (**b**), HeLa (**c**), and HepG2 (**d**) cells after 24 h treatments with ethanolic extracts from pomegranate barks (PBE) and peels (PPE).

**Table 1 plants-11-02258-t001:** Total phenolic content, flavonoid content, and antioxidant activity determined via DPPH^•^ scavenging assay of ethanolic extracts from pomegranate barks (PBE) and peels (PPE).

Samples	Total Phenolic Content	Flavonoid Content	DPPH^•^ Assay
	(mg GAE/g sample)	(mg QE/g sample)	IC_50_ (µg/mL)
PBE	574.64	52.98	4.10
PPE	242.60	23.08	9.60

GAE, equivalent gallic acid; QE, equivalent quercetin; BHT, butylated hydroxytoluene; IC_50_, the half maximal inhibitory concentration.

**Table 2 plants-11-02258-t002:** Chemical composition of the ethanolic extracts from pomegranate barks.

No.	Retention Time (min)	Compound	Peak Area (%)	Similarity Index (%)
1	5.15	Decane	9.53	76
2	5.24	Cyclotetrasiloxane, octamethyl-	9.43	86
3	6.54	Undecane	8.26	93
4	7.31	Cyclopentasiloxane, decamethyl-	4.00	78
5	8.53	5-Hydroxymethylfurfural	23.76	93
6	9.41	Phenol, 2-methyl-5-(1-methylethyl)-	15.31	90
7	9.64	Cyclohexasiloxane, dodecamethyl-	5.59	72
8	10.07	Pseudopelletierine	3.53	72
9	12.05	2,4-Di-tert-butylphenol	11.91	95
10	16.48	Hexadecanoic acid, methyl ester	6.51	93
		Total	97.83	

**Table 3 plants-11-02258-t003:** Chemical composition of the ethanolic extracts from pomegranate peels.

No.	Retention Time (min)	Compound	Peak Area (%)	Similarity Index (%)
1	3.70	Furfural	8.13	72
2	4.18	4-Cyclopentene-1,3-dione	3.83	72
3	5.27	Cyclotetrasiloxane, octamethyl-	6.86	86
4	7.35	Cyclopentasiloxane, decamethyl-	5.33	91
5	8.50	5-Hydroxymethylfurfural	33.19	93
6	8.84	Benzene, 1,3-bis(1,1-dimethylethyl)-	8.40	74
7	9.46	Phenol, 2-methyl-5-(1-methylethyl)-	12.62	90
8	9.69	Cyclohexasiloxane, dodecamethyl-	6.98	87
9	12.10	2,4-Di-tert-butylphenol	6.09	95
10	16.52	Hexadecanoic acid, methyl ester	4.80	98
		Total	96.23	

**Table 4 plants-11-02258-t004:** Minimum inhibitory concentration (MIC) of ethanolic extracts from pomegranate barks (PBE) and peels (PPE).

Bacterial Strains	PBE	PPE
	MIC (mg/mL)	MIC (mg/mL)
*E. coli* ATCC 25922	6.25	25
*S. aureus* ATCC 29213	1.56	6.25
*S.* Enteritidis ATCC 13076	1.56	3.125
*S.* Typhimurium ATCC 14028.	3.125	3.125

**Table 5 plants-11-02258-t005:** The half maximal inhibitory concentration (IC_50_) of ethanolic extracts from pomegranate barks (PBE) and peels (PPE) on the proliferation of HeLa and HepG2 cells.

Plant Extracts	IC_50_ Value (µg/mL)			
	HeLa		HepG2	
	MTT Assay	SRB Assay	MTT Assay	SRB Assay
PBE	57.5	52.8	18.6	15.9
PPE	63.2	71.4	19.5	24.5

## Data Availability

The data presented in this study are available within the article.

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
