# Peer review of "Natural Antioxidant, Antibacterial, and Antiproliferative Activities of Ethanolic Extracts from Punica granatum L. Tree Barks Mediated by Extracellular Signal-Regulated Kinase"

_plants, 2022, doi:10.3390/plants11172258_

Round 1

Reviewer 1 Report

This paper is a case study aiming to report some findings related with  the antioxidative, antibacterial and antiproliferative effects of the ethanolic extracts obtained from pomegranate barks (PBE) and of pomegranate peels (PPE); total phenolic content (TPC), total flavonoid content (TFC) and antioxidant activity (DPPH) were determined.

 From the methodological point of view, the degree of novelty for this approach is low; the applied assays (TPC, TFC, TAC, DPPH) are also quite old. However, it is possible that certain findings related with the biological activity of the extracts can be of interest.

 Unfortunately the manuscript cannot be published in this stage. Please consider the following suggestions for an improved version:

-          English needs improvement, as well as the technical language

-          #2.1 – do not duplicate in the manuscript the results reported in the table 1 (L.85-91); discuss them

-          L.92 - since the anthocyanins were not measurable, it is useless to mention them in this study (filler!), hence consider removing all related issues (L.92, 168-169, 171-176, 195, 261-271, 441, 482-483

-          #Table 1 – why are vitamin C, vitamin E and BHT mentioned in this table? It would be better to give the values for the fruit, to have a reference for how valuable are the stated by products in comparison with the edible part of the fruit

-          L.101 – if you mention that those compounds were determined, you have to give their concentrations, otherwise you have to specify that they were only identified; 23.76 and 33.19 are not concentrations, but peak areas!!!

-          #Table 2 and Table 3 – please define what is the “similarity index” and consider reviewing the names of the reported compounds

-          #2.1 – consider adding representative GC chromatograms

-          L.149 – 2.5 should be 2.6

-          L.198-200 – forced conclusion; you have to prove that 5-hydroxymethylfurfural is responsible for the reported biological effects, not to “suggest” that because this compound was one of the major ones detected by GC analysis, it is the one!

-          #4 – missing “materials” section – add the reagents and materials used in this study, together with their characteristics and providers

-          #4.1 give relevant data related with extraction (grinding method & equipment, amounts of plant materials subjected to extraction, volumes of ethanol used, volumes of extracts subjected to lyophilization, final volumes of extracts used in experiments); without these, the whole work is speculative!

-          L.246-247 – re-phrase “Folin–Ciocalteu reagent” – in this context should be a method

-          L.248 – do mention the “known dilution”

-          L.250 – what is RT?

-          L.251 – add the type and producer for the spectrophotometer

-          L.252 – add the range for the calibration

-          L.255 – do mention the “known dilution”

-          L.260 – add the type and producer for the spectrophotometer

-          L.252 – add the compound used as a reference and the range for the calibration

-          L.276 – do mention those “various concentrations”

-          L.278 – add the type and producer for the spectrophotometer

-          L.286 – add dilution

-          L.294 – add relevant details about GC’s calibration

-          In #Reference – re-number this section and update the citations in text accordingly

Reviewer 2 Report

This manuscript is about functional properties of of bioactive compounds from P. granatum extracts. The presented information is poorly discussed and weakly presented. It should improve several aspects to be considered for publication in this prestigious journal.

Use DPPH instead of DPPH

Functional properties as antioxidant and antibacterial activities must be included in the introduction section, where applicate (at least for PPE).

Lines 60, 162, 164 and 367: flavonoids are phenolic-type compounds

In Tables 2 and 3: Why the similarity index in under consideration? Why is this information included?

From my viewpoint, results and discussion can be presented together in order to make easer for readers the compression of this paper.

162 – 176: Discussion on total phenolic and flavonoid content is poor and weak, discussion and comparison are not the same. Then this should be improved.

In this study, anthocyanins were not detected; however, antioxidant properties of these molecules are discussed. Why?

Deep discussion on the polyphenols as antioxidant agents is required.

Description and deep discussion on the GC – MS profile is missing, because the scope of this prestigious Journal

Lines 188 – 190: How do they act as antibacterial agents?

Lines 198 – 200: Regarding the last question. Phenolic compounds, 5-hydroximethylfurfural or together are responsible for the activities? How, explain. (see studies about synergism)

Line 367: Since the reactions involving antioxidant activity are complex, they should not be judged by a single method or redundant methods (DPPH• and ABTS•+ scavenging tests). Then, other methods based on different mechanisms such as chelating activity and lipid peroxidation inhibition properties must be evaluated to use the term antioxidant activity.

Round 2

Reviewer 2 Report

All the suggestions were well attended by the authors. However, from my viewpoint, the term “antioxidant activity” should be changed for “free radical scavenging”